# Abdominal imaging associates body composition with COVID-19 severity

**Nicolas Basty**[1]ʘ*, **Elena P. Sorokin**[2]ʘ, **Marjola Thanaj**[1], **Ramprakash Srinivasan**[2], **Brandon Whitcher**[1], **Jimmy D. Bell**[1], **Madeleine Cule**[2], **E. Louise Thomas**[1]*

**1** Research Centre for Optimal Health, School of Life Sciences, University of Westminster, London, United Kingdom, **2** Calico Life Sciences LLC, South San Francisco, California, United States of America

ʘ These authors contributed equally to this work.
* L.thomas3@westminster.ac.uk (ELT); n.basty@westminster.ac.uk (NB)

**Data Availability Statement:** While the UKBiobank rules do not allow us to share Biobank data directly, the data we generated in the preparation of this manuscript has been returned to the UK

## Abstract

The main drivers of COVID-19 disease severity and the impact of COVID-19 on long-term health after recovery are yet to be fully understood. Medical imaging studies investigating COVID-19 to date have mostly been limited to small datasets and post-hoc analyses of severe cases. The UK Biobank recruited recovered SARS-CoV-2 positive individuals (n = 967) and matched controls (n = 913) who were extensively imaged prior to the pandemic and underwent follow-up scanning. In this study, we investigated longitudinal changes in body composition, as well as the associations of pre-pandemic image-derived phenotypes with COVID-19 severity. Our longitudinal analysis, in a population of mostly mild cases, associated a decrease in lung volume with SARS-CoV-2 positivity. We also observed that increased visceral adipose tissue and liver fat, and reduced muscle volume, prior to COVID-19, were associated with COVID-19 disease severity. Finally, we trained a machine classifier with demographic, anthropometric and imaging traits, and showed that visceral fat, liver fat and muscle volume have prognostic value for COVID-19 disease severity beyond the standard demographic and anthropometric measurements. This combination of image-derived phenotypes from abdominal MRI scans and ensemble learning to predict risk may have future clinical utility in identifying populations at-risk for a severe COVID-19 outcome.

## Introduction

COVID-19, the disease caused by the virus SARS-CoV-2 (severe acute respiratory syndrome coronavirus 2), impacts human health and has debilitating effects in both the short- and longer term [1]. A complex array of sequelae associated with COVID-19 have been reported [2], the risks of which increase in tandem with disease severity [3]. The commonly observed COVID-19 hyperinflammatory response results in damage to multiple systems and organs [4]. Individuals with both advanced age as well as chronic comorbidities such as obesity, impaired metabolic health, and diabetes are known to be at higher risk of severe disease and poorer prognosis following COVID-19 infection [5–7], and potentially at greater risk of more severe

Biobank and is available on request specifying the following field IDs.https://biobank.ndph.ox.ac.uk/showcase/label.cgi?id=158 For example, Spleen volume data is field 21083.

**Funding:** This research was funded by Calico Life Sciences LLC. https://www.calicolabs.com. The funders had no role in study design, data collection and analysis, decision to publish, or preparation of the manuscript.

**Competing interests:** I have read the journal's policy and the authors of this manuscript have the following competing interests: EPS, RS, and MC are employees of Calico Life Sciences LLC. NB, BW, JDB, MT, and ELT have no competing interests.

infection since obesity and impaired metabolic health have been associated with vaccine-breakthrough [8].

Whilst there are few imaging studies that prospectively delineate the impact of COVID-19 on organ health, there are multiple case reports and imaging studies using computed tomography (CT), magnetic resonance imaging (MRI), and ultrasound to assess patients post-infection, with some showing consistent changes characteristic of the response to COVID-19 infection. These include cardiac changes in inflammatory edema, fibrosis, impaired ventricular function, and mass [9–12], with acute myocardial injury and myocarditis reported in 8–12% patients discharged post COVID-19 and consequently increased risk of heart failure and cardiac arrhythmias [13–15]. Characteristic damage to the lungs such as the widely reported "ground-glass opacification" due to interstitial thickening, parenchymal abnormalities, edema, and reduced vital capacity, have also been reported [11,12,16]. A significant proportion of patients hospitalized with COVID-19 have developed moderate or severe kidney damage, resulting in dialysis [17–19], with reported imaging findings including renal infarcts, increased cortical echogenicity (a marker of renal disease), increased T1 (reported to indicate declining kidney function), reduced renal perfusion and increased perinephric fat stranding [11,12,20,21]. Reported changes in the liver include inflammation, fibrosis, elasticity, with conflicting reports regarding changes in liver fat and the biliary system including cholangiopathy, and presence of gallbladder 'sludge' [12,22–25]. Despite suggestions of increased risk of diabetes post infection [26], as well as biochemical changes suggesting pancreatic involvement, imaging observations have been few, with reports of a 'bulky' pancreas and peripancreatic fat stranding and inflammation [12,27]. Muscle involvement is frequently reported, from acute imaging observations, such as edema and myositis, which appear to be more common in patients with myalgia and long-term muscular changes such as sarcopenia and cachexia [28]. However, it is difficult to definitively discern from many of these studies whether the findings arise directly from SARS-CoV-2 infection, relate to the demographics of the patient population, or are in part related to the actual clinical treatment, including mechanical ventilation/hypoxia, effects of systemic inflammation, drug toxicity or changes relating to prolonged inactivity [28,29].

The UK Biobank (UKBB) is a prospective cohort study of half a million adults in the UK. Beginning in 2014, the UKBB implemented an extensive standardized imaging protocol covering the abdomen, heart and brain [30]. To investigate longitudinal changes attributable to infection with SARS-CoV-2, the UKBB invited individuals who had recovered from SARS-CoV-2 and individuals who were matched negative controls to attend a re-imaging session to determine the impact of COVID-19 on body composition and organ health. The UKBB has already provided invaluable insights regarding the prevalence of COVID-19 [31], as well as its associations with cardiometabolic profiles [32], frailty [33], and the ability to predict disease severity [34,35] and mortality [36]. In contrast to the prevailing literature that COVID-19 can result in a number of cardiovascular disorders [37], an analysis of cardiac MRI found no significant changes between cases before and after infection, compared to controls, which the authors attributed to generally a milder disease in the UKBB cohort compared to most clinical studies [38]. However, a brain MRI study from the same UKBB cohort reported multiple changes including loss of gray matter in several regions of the brain associated with the primary olfactory and gustatory systems [39].

The aim of the current study is to assess the impact of SARS-CoV-2 infection on abdominal organ health and body composition in 967 cases and 913 controls. Moreover, we assessed the association between severity of COVID-19 disease and multi-organ image-derived phenotypes from the baseline imaging visit. The findings from this study are relevant for determining

long-term outcomes and future health needs for both populations and individuals recovering from COVID-19, as well as shedding light on possible factors for disease severity.

## Materials and methods

### Study design

Approximately 45,000 UKBB participants have attended a baseline MRI scanning session (brain, heart, and abdomen) prior to the appearance of SARS-CoV-2 in the UK. Starting from February 2021, a total of 1,880 participants who attended the first imaging visit were recruited to a new re-imaging study aiming to determine the impact of COVID-19. Individuals with a confirmed COVID-19 diagnosis (from results available through primary care data, hospital records, antigen tests obtained from Public Health data records, or home-based antibody lateral flow kits sent by the UKBB to participants) were invited to take part in the study, together with a matched control group. The control group was selected based on negative results or no history of positive results from the aforementioned sources. Cases and controls were paired based on sex, ethnicity, age (± 6 months), imaging assessment centre, and date of initial baseline scan (± 6 months). Due to small numbers of non-white participants, ethnic matching was based on a classification of white vs non-white. In line with UK travel restrictions at the time, participants were restricted to those living within 60 km of the UKBB scanning centers at Stockport, Newcastle, and Reading. A detailed description is given here: https://biobank.ndph. ox.ac.uk/showcase/showcase/docs/casecontrol_covidimaging.pdf.

Participant data from the UKBB cohort was obtained as previously described [40] through UKBB Access Application number 44584. The UKBB has approval from the North West Multi-Centre Research Ethics Committee (REC reference: 11/NW/0382). All measurements were obtained under these ethics, adhering to relevant guidelines and regulations, and written informed consent was obtained from all participants. Researchers may apply to use the UKBB data resource by submitting a health-related research proposal that is in the public interest. More information may be found on the UKBB researchers and resource catalogue pages (https://www.ukbiobank.ac.uk).

### Image-derived phenotypes

Full details regarding the UKBB MRI abdominal protocol have previously been reported [30]. The data included in this paper focused on the neck-to-knee Dixon MRI acquisition, separate single-slice quantitative MRI acquisitions of the liver and pancreas, and T1-weighted pancreas volume. We processed all available image data for cases and controls using our previously published image processing and segmentation pipelines [41]. We subsequently generated image-derived phenotypes (IDPs) of abdominal organs, adipose tissue, and muscles using convolutional neural networks [41–43]. As in [41] we performed validation procedures that included a visual inspection in three separate groups of scans: the smallest fifty by volume, the largest fifty by volume and fifty randomly-selected scans not included in the previous two groups. We also inspected the segmentations with the largest variations between imaging visits. Evaluating the extreme cases provided insight into our ability to process the full spectrum of organ shapes and sizes, while the randomly-selected scans provided confidence in the vast majority of scans, in addition to high image segmentation metrics on out-of-sample data. *During visual quality control of the segmentations, no catastrophic failures were observed and we did not perform any manual editing. We defined a catastrophic failure to be when segmented organs had major components missing or components that did not belong to the structure added (i.e. over or under segmentation). This could be due to underlying data errors such as fat-water swaps, or where the neural network did or did not highlight part of the organ even though the underlying image*

*intensities are free of ambiguities*. We included a total of twelve IDPs in this study: volumes of abdominal subcutaneous adipose tissue (ASAT), visceral adipose tissue (VAT), liver, lungs, iliopsoas muscles, kidneys, pancreas, spleen, as well as proton density fat fraction (PDFF) measures of liver and pancreas fat content, and organ iron concentration of the liver and pancreas.

## Statistical analysis of IDP data

All summary statistics, hypothesis tests, models were performed using the R3.6.3 software environment for statistical computing and graphics [44]. Visualization was performed using the *ggplot2* v3.3.5 package. Descriptive statistics are provided as mean, standard deviation, and range for continuous traits, and as mean, standard deviation, and 95% confidence interval for binary traits. Differences between groups were assessed for statistical significance using a Chi-squared goodness-of-fit test for binary traits, and Student's t-test for continuous traits.

In our analyses, we assessed the longitudinal effects of SARS-CoV-2 infection as well as the severity of COVID-19 based on pre-pandemic imaging alone.

## Longitudinal analysis

Associations between changes in imaging phenotypes and COVID-19 diagnosis required calculating the age difference between the two imaging visits: $\Delta age = (age_{rescan} - age_{baseline})$. To capture quadratic effects of age, $\Delta age^2$ was calculated as the square of the difference of the age at rescanning and the age at baseline: $\Delta age^2 = (age_{rescan} - age_{baseline})^2$. The following model was selected to associate changes in abdominal IDPs with COVID-19 diagnosis:

$$IDP_{rescan} = X\beta + IDP_{baseline}\beta_{IDP} + \Delta age \cdot \beta_{age} + \Delta age^2 \cdot \beta_{age^2} + COVID \cdot \beta_{COVID} \quad (1)$$

*X* describes fixed effects: sex, ethnicity, height, BMI, smoking status, alcohol consumption, and Townsend deprivation index. Baseline and rescan IDP values were standardized. During model selection, we initially included an interaction term between COVID positivity and sex but did not observe any significant interactions (p>0.05), so the interaction term was dropped from the final model above.

## Severity analysis using baseline data only

We determined COVID-19 severity based upon hospitalization with International Classification of Diseases 10th Revision (ICD-10) diagnosis codes U071.1 or U071.2, plus a positive test status from one of the available sources. We also filtered severe cases by pneumonia diagnosis (J128.2) and by placement on a ventilator (Z99.11). For the severity analyses of first imaging visit data only, a second category of COVID-19 severe outcomes also included death with a recorded cause of death as COVID-19 (U07.1). Follow-up analysis of pulmonary diseases was conducted using hospital billing codes J00-J99. The severity analysis included 140 additional SARS-CoV-2 positive samples beyond the matched case/control design in the first analysis, of participants who had attended the first imaging visit but were not recruited as part of the re-imaging study.

As a logistic regression model, COVID-19 severity was regressed on standardized baseline IDP values and fixed effect covariates:

$$severity = X\beta + IDP_{baseline} \cdot \beta_{IDP} \quad (2)$$

Here, *X* included rescan imaging age, imaging age squared, sex, ethnicity, Townsend deprivation index, height, BMI, smoking status, alcohol consumption, and baseline imaging center location. Baseline imaging center was encoded as a categorical variable. During model

selection we tested whether baseline IDP association with severity differed by sex but the interaction term was not significant for any of the baseline IDPs (p>0.05). We modeled severity as hospitalized and non-hospitalized outcomes; severe (hospitalization + death) and non-severe outcomes; and death vs hospitalization. For multiple test correction of independent traits, a Bonferroni adjustment was used to determine a significance threshold, where alpha was set at 0.05. For dependent traits, false discovery rate (FDR) was estimated from p-values using the Benjamini-Hochberg method [45], and a threshold of FDR ≤ 0.05 determined significance.

A multivariate model was also developed for COVID-19 severity with a penalty term added to the least-squares loss function to implement LASSO L1-regularization.

$$L(x) = \sum_{i=1}^{n}(Y_i - \sum_{j=1}^{p} X_{ij}\beta_j)^2 + \lambda\sum_{j=1}^{p}|\beta_j| \tag{3}$$

Here, $X$ included twelve IDPs adjusted for covariates. LASSO was implemented using the *glmnet* package. The shrinkage parameter $\lambda$ was set as 0.0238, its minimum value during cross-validation.

We developed random forest [46] classifiers for COVID-19 severity prediction in python *3.7.2* using the *scikit-learn 1.0.2* package [47]. We tested an increasing number of forests, doubling the estimator number from 1 to 1024, and found the best results with 128 trees. Model training was conducted with 10-fold cross-validation on an 80% randomized data split for training, with the remaining 20% kept aside as a testing set used to assess model performance. Area under the receiver operating characteristic (ROC) curve (AUC) and F1 score were used as performance metrics. We trained three kinds of models, two without IDPs and one with IDP, in order to assess potentially added value from IDPs on top of standard anthropometric traits.

## Results

An example of the 3D segmentations obtained from the neck-to-knee Dixon MRI acquisition for one of the COVID-19 rescan participants (Fig 1) (generated using 3D Slicer [48]), including the liver (yellow), lungs (blue), spleen (purple), kidneys (green), ASAT (white), VAT (orange transparent, making internal organs visible), iliopsoas muscles (pink), and the pancreas (red).

Of 1,955 matched cases and controls recruited in the COVID-19 study, 51.2% (n = 1,000) tested positive for SARS-CoV-2 via one of four assays, and 48.8% (n = 955) tested negative. We examined baseline demographics of the COVID-19 study cohort with complete covariates (n = 1,880) including age, sex, ethnicity, height, BMI, waist/hip ratio, blood pressure, Townsend deprivation index, smoking status, and diagnosis rate of several common diseases. For all demographic parameters, the differences did not achieve the threshold for statistical significance, indicating effective case-control matching (Table 1).

To identify changes in abdominal IDPs associated with SARS-CoV-2 test positivity, we first examined the distributions in abdominal IDPs between the baseline scan and repeat scan. We observed differences in the distributions of abdominal IDPs by sex (Table 2). To test whether SARS-CoV-2 diagnosis was a significant factor in abdominal IDP changes between the two imaging visits, we developed a multiple linear regression model, adjusting for sex and additional confounding variables (Methods). Sex, age, and BMI were among the demographic parameters most highly associated with changes in abdominal IDPs between baseline and rescan visits (Fig 2 and S1 Table). SARS-CoV-2 diagnosis was associated with a decrease in lung volume (beta = -0.07, FDR = 0.02) in a regression model, even after adjusting for delta age, delta age squared, sex, ethnicity, height, BMI, smoking status, alcohol consumption, and Townsend deprivation index (Fig 2 and S1 Table). Consistent with this observation,

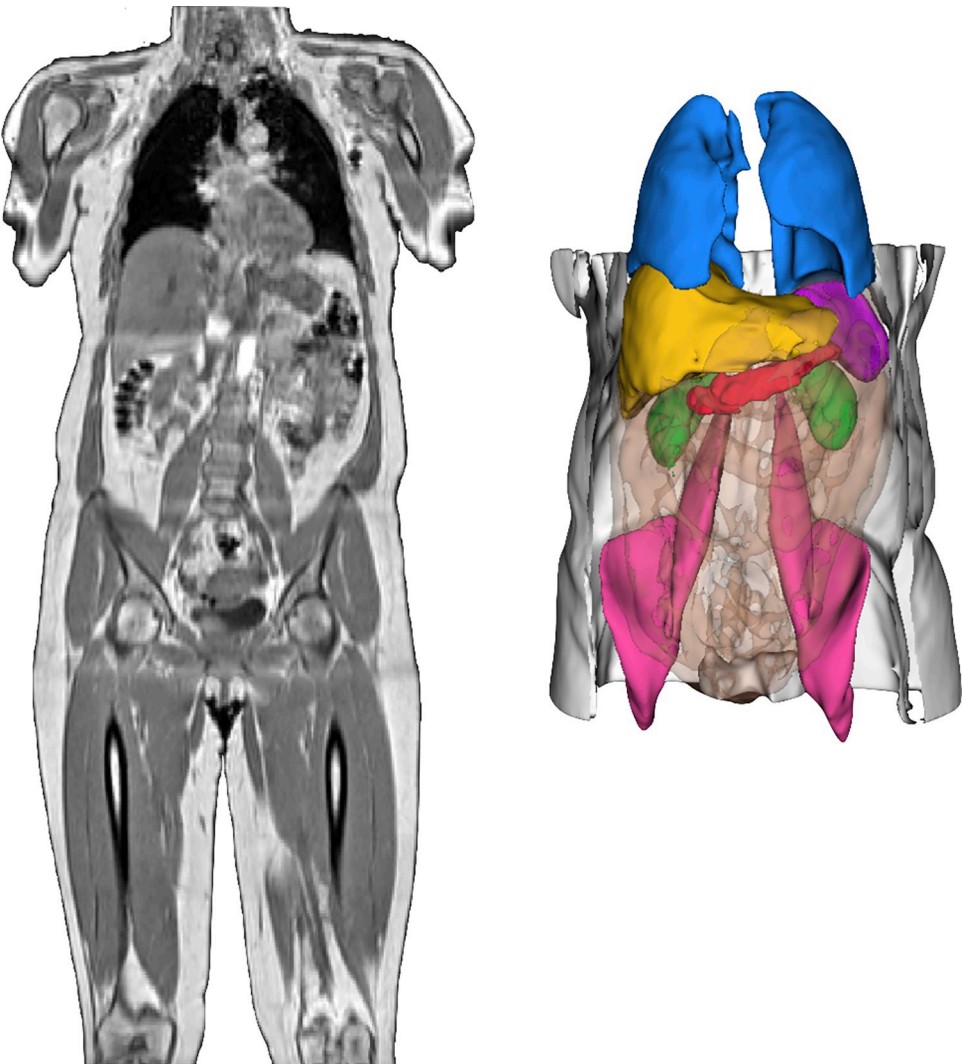

**Fig 1. Neck-to-knee UK Biobank MRI acquisition of a COVID-19 study participant and 3D renderings of image-derived phenotypes.** These were obtained after image preprocessing and segmentation pipelines for that same participant. 3D segmentations: Liver (yellow), lungs (blue), spleen (purple), kidneys (green), abdominal subcutaneous adipose tissue (white), visceral adipose tissue (orange transparent), iliopsoas muscles (pink), pancreas (red).

pulmonary disease diagnoses during the study period were enriched in SARS-CoV-2 positive cases (n = 158) versus in the matched controls (n = 117) ($p_{chisq}$ = 0.03).

We tested whether baseline IDP values, assessed before the pandemic, were associated with severity of COVID-19 disease. We defined severity based on first ascertaining for SARS-CoV-2 test positivity during the study period, then combining death, hospitalization, and GP records, including searching for hospitalization with COVID-19, critical care treatment, pneumonia diagnosis, and/or placement on a ventilator during the study period. Of the baseline imaging cohort with a positive SARS-CoV-2 test result included in this study (n = 1,107), 16.1% were accompanied by severe disease, comprising hospitalized cases (n = 149) and deaths (n = 30) from COVID-19. The remaining participants who tested positive for SARS-CoV-2 in this study did not have severe disease (n = 928).

In univariate regression modeling of hospitalization versus death from COVID-19, the group that died was significantly older than the hospitalized group (mean difference = 4.7

**Table 1. Baseline population demographics of the UK Biobank COVID-19 study cohort (n = 913 controls and n = 967 cases).**

|  | Controls (n = 913) | Cases (n = 967) | p value |
|---|---|---|---|
| Age at scan 1 (years) | 60.3 (7.5) [46–80] | 60 (7.6) [46–81] | 0.427 |
| Age at scan 2 (years) | 63.5 (7.3) [50.6–82.1] | 63.2 (7.2) [50.4–82.7] | 0.418 |
| Female (%) | 52 | 55 | 0.389 |
| White (%) | 83 | 84 | 0.569 |
| Height (cm) | 169.8 (9.3) [145–198] | 169.7 (9.1) [146–204] | 0.923 |
| BMI (kg/m2) | 26.4 (4.4) [16.6–45.1] | 26.7 (4.5) [17.4–51.5] | 0.262 |
| Waist/Hip Ratio | 0.9 (0.1) [0.6–1.1] | 0.8 (0.1) [0.6–1.1] | 0.318 |
| Systolic blood pressure (mmHg) | 131.7 (16.6) [95–204.5] | 131.3 (17) [91.5–216] | 0.578 |
| Diastolic blood pressure (mmHg) | 80.9 (9.8) [56–112.5] | 80.4 (10) [56.5–124] | 0.324 |
| Townsend index | -1.7 (2.7) [-6.2–8.1] | -1.5 (2.8) [-6.3–8.6] | 0.166 |
| Smoker (%) | 6 | 6 | 0.584 |
| Alcohol drinker (%) | 98 | 98 | 0.352 |
| Asthma (%) | 13 | 13 | 0.888 |
| COPD (%) | 1 | 1 | 0.911 |
| Myocardial infarction (%) | 2 | 2 | 0.861 |
| Stroke (%) | 0 | 0 | 1.000 |
| Cancer (%) | 11 | 12 | 0.563 |
| T1D (%) | 0 | 0 | 1.000 |
| T2D (%) | 3 | 4 | 0.342 |

For continuous variables, mean, standard deviation, and range are shown. COPD, chronic obstructive pulmonary disease. BP, blood pressure. Unadjusted p-values are shown from Student's t-test or Chi-squared goodness-of-fit tests respectively for continuous and binary variables. P-values with an asterisk (*) are statistically significant with FDR ≤0.05.

years; FDR = 0.008) (Table 3). In univariate models of all severe cases (n = 179) tested against non-severe SARS-CoV-2 infections (n = 928), we found that increased age, male sex, increased BMI, increased waist/hip ratio, increased blood pressure, being a smoker, chronic obstructive pulmonary disease (COPD), stroke, and myocardial infarctio were all associated with increased risk of having a severe outcome (FDR < = 0.05) (Table 3).

We observed differences in the distributions of baseline abdominal IDPs by disease severity after stratifying by sex (Table 4). To test whether baseline abdominal IDPs were associated with COVID-19 severity, we performed multiple linear regression and found that sex and BMI were predictors of COVID-19 severity (Fig 3 and S2 Table). We tested whether any of the twelve IDPs were also associated with any severe outcome in the model and found that iliopsoas muscle volume was negatively associated with severity (beta = -0.68; FDR = 0.0057), while two variables were positively associated with severity: VAT volume (beta = 0.42; FDR = 0.0036) and liver PDFF (beta = 0.26; FDR = 0.043), even after adjustment for ten possible confounders. When testing a milder severity score measured by hospitalization vs non-hospitalization only, iliopsoas muscle volume was associated (beta = -0.65, FDR = 0.012) (S3 Table). In a multivariate model of association for COVID-19 severity performed with L1 penalization, visceral fat was significant (beta = 2.2e-4, p = 4.9e-4) (S4 Table).

We tested whether IDPs could be used to classify and predict COVID-19 disease severity using machine learning with random forests. We compared the performance of a base demographic model (age, sex) to an anthropometric model (age, sex, height, BMI), and finally a full model containing IDPs in addition to demographic and anthropometric predictors. Age and

**Table 2. Summary data on image-derived phenotypes for n = 913 SARS-CoV-2 positive cases and n = 967 SARS-CoV-2 matched controls, stratified by imaging visit and sex.**

| Sex | Female | | | | Male | | | |
|---|---|---|---|---|---|---|---|---|
| Scan | Baseline | | Reimaging | | Baseline | | Reimaging | |
| Phenotype | Controls (n = 478) | Cases (n = 526) | Controls (n = 478) | Cases (n = 526) | Controls (n = 435) | Cases (n = 441) | Controls (n = 435) | Cases (n = 441) |
| ASAT volume (ml) | 9727 (4666) | 9921 (4638) | 9822 (4741) | 10043 (4664) | 6933 (3004) | 7261 (3241) | 6858 (2932) | 7446 (3339) |
| Iliopsoas muscle (ml) | 262 (41) | 264 (38) | 257 (40) | 259 (37) | 403 (69) | 406 (63) | 392 (68) | 394 (62) |
| Kidney volume (ml) | 132 (25) | 133 (24) | 123 (23) | 125 (22) | 161 (29) | 163 (27) | 148 (28) | 150 (25) |
| Liver iron (mg/g) | 1.18 (0.2) | 1.22 (0.28) | 1.21 (0.2) | 1.23 (0.42) | 1.22 (0.28) | 1.22 (0.27) | 1.25 (0.23) | 1.24 (0.23) |
| Liver PDFF (%) | 4.06 (4.25) | 3.76 (4.06) | 4.79 (5.31) | 4.21 (4.12) | 5.33 (4.7) | 6.04 (6) | 5.99 (5.79) | 6.71 (6.03) |
| Liver volume (ml) | 1329 (264) | 1334 (233) | 1314 (293) | 1327 (253) | 1553 (288) | 1581 (317) | 1509 (300) | 1556 (325) |
| Lung volume (ml) | 2241 (507) | 2271 (569) | 2289 (523) | 2276 (539) | 2966 (782) | 2876 (762) | 3000 (713) | 2866 (733) |
| Pancreas iron (mg/g) | 0.79 (0.1) | 0.79 (0.08) | 0.79 (0.08) | 0.78 (0.08) | 0.75 (0.09) | 0.74 (0.07) | 0.75 (0.07) | 0.75 (0.06) |
| Pancreas PDFF (%) | 7.15 (5.75) | 7.58 (6.28) | 8.04 (6.57) | 7.69 (6.22) | 12.05 (8.78) | 12.19 (8.02) | 13.51 (9.5) | 12.73 (8.27) |
| Pancreas volume (ml) | 59 (15) | 59 (14) | 59 (15) | 59 (14) | 67 (15) | 67 (17) | 66 (15) | 67 (17) |
| Spleen volume (ml) | 149 (51) | 156 (56) | 140 (52) | 148 (57) | 202 (70) | 205 (70) | 191 (73) | 197 (73) |
| VAT volume (ml) | 2645 (1593) | 2692 (1506) | 2772 (1644) | 2854 (1569) | 4956 (2275) | 5171 (2300) | 5030 (2397) | 5392 (2363) |

Amongst females, there were 478 controls and 526 cases. Amongst males, there were 435 controls and 441 cases. ASAT: Abdominal subcutaneous adipose tissue, VAT: Visceral adipose tissue, PDFF: Proton density fat fraction of fat content estimated from MRI and presented as a percentage. For each variable, the mean and standard deviation are shown.

sex were modestly able to discriminate between severe and non-severe outcomes (AUC = 0.67), adding height and BMI as additional predictors performed better (AUC = 0.75), and the full model was best able to predict disease severity (AUC = 0.82) (Table 5 and Fig 4A). The best-performing model contained age, sex, height, BMI as well as visceral fat, iliopsoas muscle volume, and liver fat. Investigating feature importances showed that the most significant feature contribution to the random forest classifier came from VAT (19.7%) (Fig 4B). Finally, we used the best-performing random forest classifier to predict severity in the remaining UKBB baseline imaging cohort (n = 43,464), and estimated that 2,309 individuals (5.31%) would have severe disease.

## Discussion

The UK Biobank (UKBB) spearheaded an ambitious endeavor to acquire extensive imaging data for over 100,000 subjects since 2014, and conceived the COVID-19 study to enable this unique longitudinal precision phenotyping research program. For this study, we produced and analyzed twelve abdominal IDPs for 967 cases and 913 matched controls of the UKBB longitudinal COVID-19 study, as well as another 140 severe cases of participants who attended the first imaging visit, identified from hospital records and the death register available from UKBB.

Our longitudinal analysis, examining body composition via abdominal MRI scansbefore and after COVID-19, showed a significant decrease in lung volume associated with SARS-CoV-2 positivity. The gold standard measure of lung function is spirometry, with lung volumes generally assessed using whole-body plethysmography or single-breath helium dilution [49,50]. MRI has previously been used to measure lung volume, but this generally involves acquisition of images during breath-hold at full inflation [51]. Although the neck-to-knee MRI

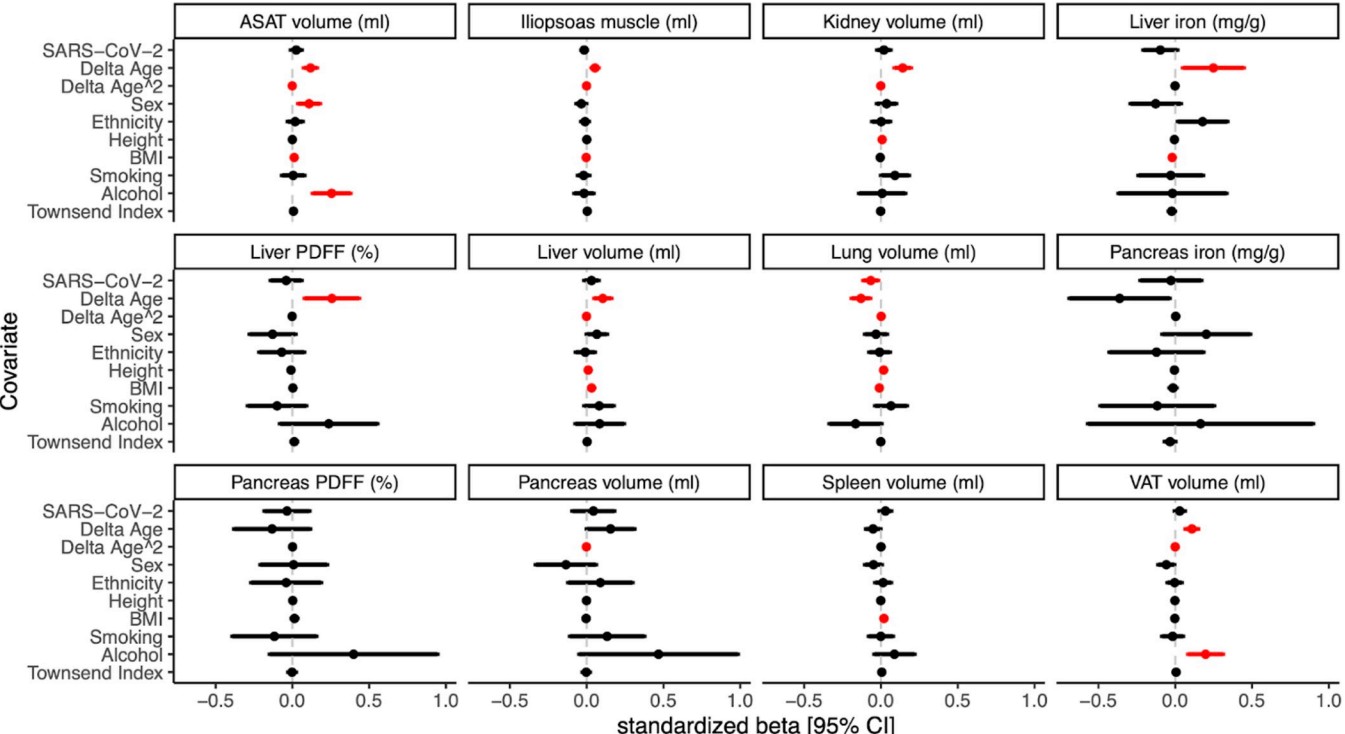

**Fig 2. Changes in abdominal image-derived phenotypes are associated with age, sex, BMI, and SARS-CoV-2 test positivity.** There were 967 positive cases and 913 matched negative controls for a total sample size of 1,880. Multiple linear regression modeling was performed (Methods). 95% confidence intervals around standardized betas are shown. Significant associations (FDR≤0.05) are shown in red.

acquisition in the UKBB is not conventionally used for measurements of lung volumes, it does allow opportunistic assessment of this organ [41]. Thus, our observation of a reduction in lung volume following COVID-19 is consistent with published studies that have shown persistent lung abnormalities following COVID-19 infection, including reduced forced expiratory volume, vital capacity, and forced vital capacity measured by spirometry [52,53]. Moreover, CT scans showing both reduced lung volume [54] and impaired functional lung volume related to disease severity [55] have been reported. Other studies using pulmonary function tests have reported that lung volumes in patients with mild/moderate COVID-19 are normal [56]. However, most of these studies lacked pre-infection assessment, whereas in the current study, pre-infection imaging information was available. Although the MRI sequences employed in this study were not designed to detect radiological changes such as the characteristic ground-glass opacification, consolidation and lesions [11,16], they may be able to provide additional insight into changes occurring as a consequence of infection with SARS-CoV-2. It is worth noting that contrary to most other imaging-based studies relating to COVID-19, which recruited predominantly hospitalized or severe cases, the UKBB longitudinal COVID-19 study deals predominantly (96% or 928 out of 967) with mild cases of the disease. It is therefore interesting that the longitudinal analysis showed significant decrease in lung volume in a population of mostly mild cases, after adjusting for possible confounding factors, providing insight into mild or asymptomatic responses to the disease.

No other significant longitudinal changes were observed in any of our twelve IDPs following COVID-19 infection, despite previous case-control studies reporting differences detectable by MRI attributed to COVID-19 in multiple organs including the liver, kidneys, pancreas,

**Table 3. Baseline demographics for all SARS-CoV-2 positive cases in the UK Biobank with baseline imaging data, stratified by severity of COVID-19 disease.**

|  | Death (n = 30) | Hospitalization (n = 149) | p | All severe (n = 179) | Not severe (n = 961) | p |
|---|---|---|---|---|---|---|
| Age at initial scan | 70.1 (5.89) [57–80] | 65.44 (8.29) [49–80] | 5.65E-04* | 66.22 (8.11) [49–80] | 59.82 (7.53) [46–81] | 2.97E-19* |
| Female (%) | 23 | 30 | 0.69 | 29 | 56 | 2.51E-11* |
| European (%) | 73 | 81 | 0.52 | 80 | 84 | 0.71 |
| Height (cm) | 171.72 (8.87) [154–190] | 171.52 (8.16) [151–190] | 0.91 | 171.55 (8.26) [151–190] | 169.67 (9.1) [146–204] | 7.24E-03* |
| BMI (kg/m2) | 28.91 (4.36) [20.74–39.23] | 29.14 (5.33) [18.37–47.58] | 0.80 | 29.11 (5.17) [18.37–47.58] | 26.54 (4.42) [17.38–51.49] | 3.88E-09* |
| Waist/Hip Ratio | 0.93 (0.08) [0.74–1.06] | 0.91 (0.09) [0.72–1.11] | 0.30 | 0.92 (0.09) [0.72–1.11] | 0.85 (0.09) [0.63–1.12] | 4.17E-20* |
| Systolic BP (mmHg) | 140.41 (17.22) [108.5–178] | 137.88 (18.68) [93.5–215] | 0.49 | 138.31 (18.41) [93.5–215] | 130.95 (16.7) [91.5–216] | 3.24E-06* |
| Diastolic BP (mmHg) | 82.39 (6.89) [68–94.5] | 83.05 (10.21) [58.5–119] | 0.67 | 82.94 (9.71) [58.5–119] | 80.26 (9.89) [56.5–124] | 1.33E-03* |
| Townsend index | -1.26 (3.49) [-5.21–6.75] | -1.38 (3.21) [-6.16–9.32] | 0.86 | -1.36 (3.25) [-6.16–9.32] | -1.46 (2.85) [-6.26–8.63] | 0.70 |
| Smoker (%) | 17 | 15 | 1.0 | 16 | 6 | 4.60E-05* |
| Alcohol consumption (%) | 100 | 95 | 0.49 | 96 | 98 | 0.41 |
| Asthma (%) | 10 | 14 | 0.76 | 13 | 13 | 0.71 |
| COPD (%) | 10 | 7 | 0.80 | 8 | 1 | 8.66E-08* |
| MI (%) | 7 | 5 | 1.0 | 5 | 2 | 1.60E-02* |
| Stroke (%) | 0 | 4 | 0.55 | 3 | 0 | 1.75E-04* |
| Cancer (%) | 17 | 15 | 1.0 | 15 | 11 | 0.15 |
| T2D (%) | 12 | 17 | 1.01 | 18 | 3 | 1.17E-05* |

Here, severity included both death (n = 30) and hospitalization with COVID-19 (n = 149), and SARS-CoV-2 positive controls (n = 928) who did not have severe disease. COPD: Chronic obstructive pulmonary disease, MI: Myocardial infarction, T2D: Type 2 diabetes. For continuous variables, mean, standard deviation, and range are shown. Test statistics reflect T statistics for continuous variables and Chi-squared statistics for binary variables. P-values with an asterisk (*) are statistically significant with FDR ≤0.05.

spleen, and muscle [11,12,22,23,28]. Similarly, there are numerous case reports of individual patients which describe changes in organs throughout the body, although these are mostly linked to severe disease, where it is not always clear whether the reported changes relate to

**Table 4. Baseline image-derived phenotypes in COVID-19 cases, stratified by disease severity, including separately death, hospitalization, and non-severe outcomes.**

|  | Female | | | Male | | |
|---|---|---|---|---|---|---|
| IDP | Non-hospitalized | Hospitalization | Death | Non-hospitalized | Hospitalization | Death |
| ASAT volume (ml) | 9819 (4538) | 12451 (5804) | 12041 (3650) | 7221 (3266) | 8892 (4447) | 8768 (3380) |
| Iliopsoas muscle (ml) | 264 (38) | 252 (44) | 214 (39) | 408 (64) | 380 (61) | 372 (63) |
| Kidney volume (ml) | 132 (24) | 136 (33) | 133 (35) | 163 (27) | 159 (33) | 153 (36) |
| Liver iron (mg/g) | 1.22 (0.28) | 1.21 (0.21) | 1.34 (0.18) | 1.21 (0.27) | 1.22 (0.23) | 1.08 (0.22) |
| Liver PDFF (%) | 3.72 (4.04) | 6.42 (5.9) | 3.66 (2.17) | 5.98 (5.98) | 8.38 (7.15) | 7.57 (5.5) |
| Liver volume (ml) | 1330 (230) | 1487 (403) | 1380 (2645) | 1582 (319) | 1625 (665) | 1497 (385) |
| Lung volume (ml) | 2271 (568) | 2338 (638) | 2047 (391) | 2898 (766) | 2750 (838) | 2829 (1048) |
| Pancreas iron (mg/g) | 0.79 (0.08) | 0.83 (0.09) | 0.78 (0.07) | 0.74 (0.07) | 0.75 (0.07) | 0.73 (0.1) |
| Pancreas PDFF (%) | 7.29 (5.72) | 14.03 (11.81) | 12.4 (9.39) | 12.07 (7.99) | 15.2 (9.75) | 15.6 (9.43) |
| Pancreas volume (ml) | 59 (15) | 57 (15) | 57 (13) | 67 (17) | 62 (17) | 56 (17) |
| Spleen volume (ml) | 155 (54) | 178 (84) | 165 (52) | 205 (70) | 201 (76) | 194 (58) |
| VAT volume (ml) | 2646 (1464) | 3710 (1748) | 3375 (1015) | 5105 (2284) | 6619 (2486) | 7225 (2595) |

There were n = 515 non-hospitalized female cases and n = 413 non-hospitalized male cases, n = 43 hospitalized female cases, n = 106 hospitalized male cases. For the death data, 7 were female and 23 were male. ASAT: Abdominal subcutaneous adipose tissue, VAT: Visceral adipose tissue, PDFF: Proton density fat fraction of fat content estimated from MRI and presented as a percentage. For each variable, the mean and standard deviation are shown.

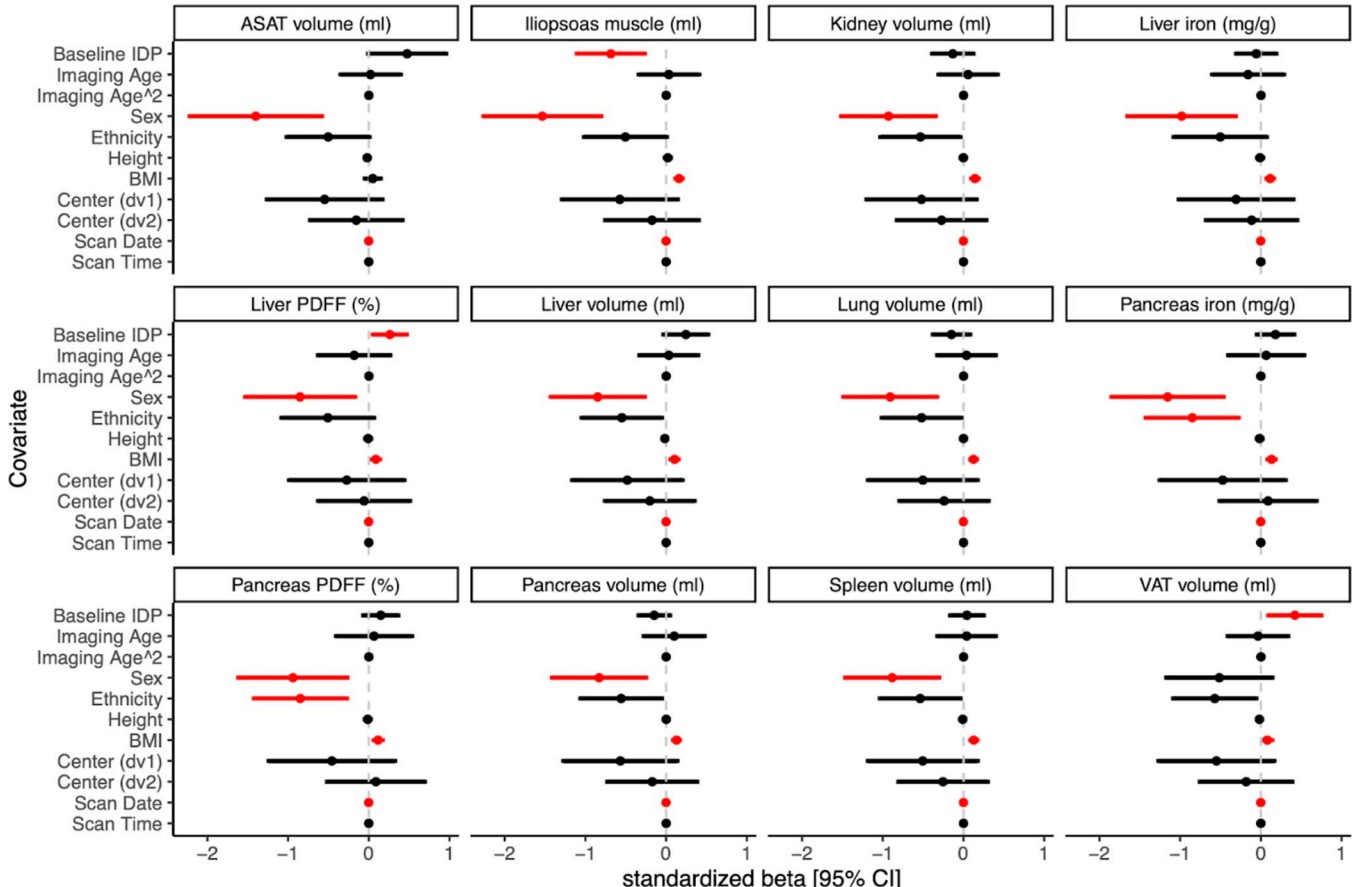

**Fig 3. COVID-19 severity and associations with baseline image-derived phenotypes.** For this regression model, severity was defined by both death (n = 30) and hospitalization (n = 149), for a total of n = 179 cases compared to n = 928 non-hospitalized COVID-19 positive controls. 95% confidence intervals around standardized betas are shown. Significant associations (FDR<0.05) are depicted in red. Some binomial variables have large confidence intervals due to class imbalance.

viral infection per se, hypoxia/mechanical ventilation or drug treatments. Interestingly, the first longitudinal study of cardiac phenotypes, also using the UKBB dedicated cardiac imaging dataset, reported no significant changes in cases vs controls, which they attributed to a generally higher prevalence of milder/non-hospitalized disease in this population [38]. However, brain MRI in the same cohort revealed significant gray matter loss in several brain areas [39], suggesting that even a non-severe COVID-19 infection can cause changes in the brain. The availability of pre-pandemic data in the current abdominal study, and also previous UKBB brain and cardiac studies, reduces the risk of misattributing effects of pre-existing conditions or risk factors to COVID-19, and enables a better insight into the disease. Interestingly, of the

**Table 5. Severity predictions using random forest.** Training was performed using random forests, considering three models.

| Model | Predictors | AUC | AUC [95% CI] | F1 score |
|---|---|---|---|---|
| Base model | age, sex | 0.686 | 0.681–0.692 | 0.124 |
| Anthropometric traits model | age, sex, BMI, height | 0.747 | 0.743–0.751 | 0.350 |
| Full model | age, sex, BMI, height, IDPs | 0.819 | 0.815–0.822 | 0.372 |

AUC: Area under the curve.

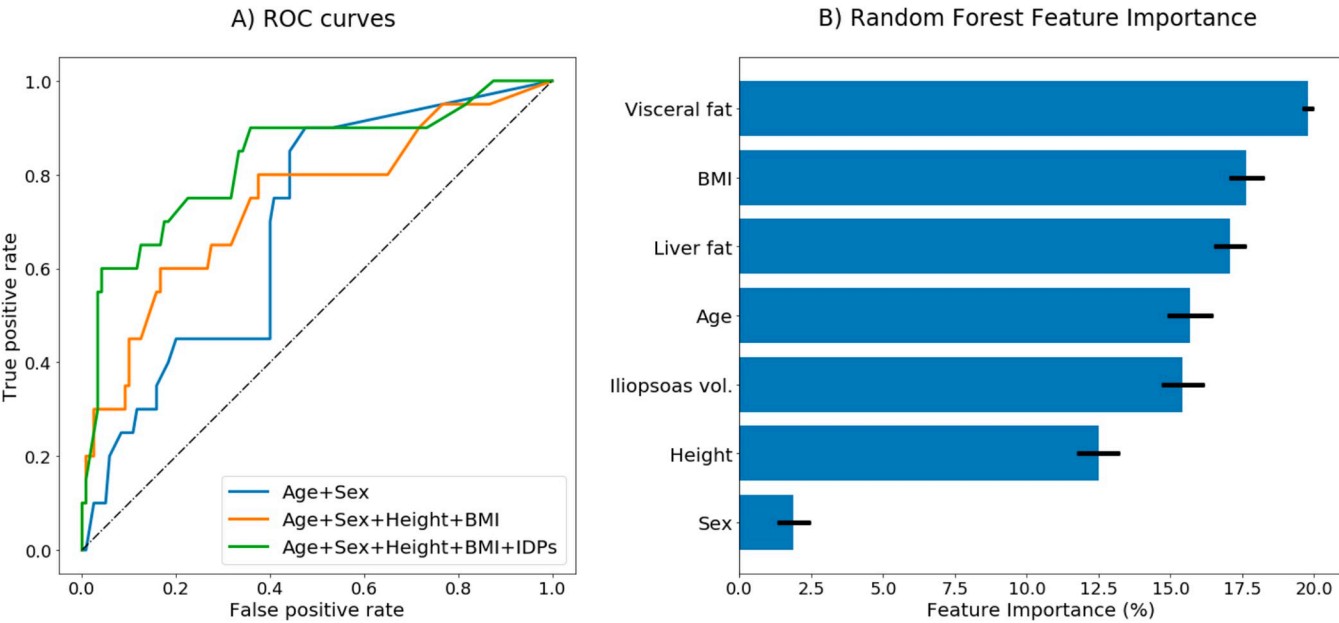

**Fig 4. Random forest severity predictions model performance.** (A) Receiver operating characteristic (ROC) curves for the three random forest models. Blue: Age+Sex, Yellow: Age+Sex+Height+BMI, Green: Age+Sex+Height+BMI+Image-Derived Phenotypes (IDPs), specifically: Visceral fat, liver fat, and iliopsoas muscle volume (B) Feature importances, plotted with error bars representing inter-tree variability for the full model (Age+Sex+Height+BMI+IDPs).

three UKBB-based COVID-19 studies, the most significant changes appear to be associated with the brain. This may in part reflect the reportedly higher susceptibility of brain tissue to inflammatory disruption [57] and/or the higher resolution of the brain MRI acquired by the UKBB which make it more likely to detect small anatomical changes.

Our analyses evaluating COVID-19 severity based on body composition prior to infection revealed that severity was associated with elevated visceral adipose tissue and liver fat content as well as smaller iliopsoas muscle volume. It should be noted that participants were not recruited for the COVID-19 re-imaging study based on disease severity, and only a small proportion of the responding participants (4% or 39 out of 967), were classified as having severe disease. Although we do not have granular information on the recruitment process, it is possible that there may have been some self-selection since there are few severe cases, and potentially severely ill participants were less inclined to attend an assessment day. However, we determined that 140 UKBB participants who had undergone a baseline imaging visit but not recruited to the re-imaging study had severe COVID-19 based on hospital records and the death register. This enabled the relationship between IDPs and disease severity to be established.

Previous studies have suggested that elevated visceral fat may affect the severity of COVID-19, associated with prognosis and requirement for intensive care [58–61]. This may be due to higher expression of angiotensin-II (*ACE2*) in visceral fat (compared with subcutaneous adipose tissue depots), resulting in elevated production of inflammatory cytokines [60]. In our random forest model, visceral fat was the most important feature contributing to disease severity. Like visceral fat, liver fat is known to be an independent risk factor for several conditions, including type 2 diabetes and cardiometabolic disease. More recently, elevated liver fat has been shown to be more prevalent in COVID-19 patients and also indicative of severity and length of hospitalization [62–64]. Our results confirm these findings, with cases with elevated liver fat showing more severe response to COVID-19 infection, as well as the third most

important feature picked up by the random forest disease severity model being liver fat. The mechanism(s) underpinning this effect is unclear, but may entail several interlinked factors, including an intensification of the "cytokine storm" by the pro-inflammatory state of fatty liver [65], a reduction in an individuals' vitamin D levels, which would make subjects more susceptible to organ damage [66], enhancement of viral replication arising from increased levels of ACE2 receptor observed in hepatic steatosis [67] and/or the fact that viral replication may be utilizing intracellular lipid stores for its propagation, making tissues with elevated levels of lipid droplets more amenable to viral damage.

Out of all of the 1,107 positive individuals with first imaging visit data, 565 (51%) were female and 542 (49%) were male. It is worth noting that of the 179 severe cases, 72.1% were male and 27.9% were female. This means that men account for more than twice as many severe cases than women in our study even though the case population is evenly distributed between men and women, which matches previous reports of male sex being a risk factor of severe illness and death from COVID-19 [68]. Men are known to accumulate more visceral fat than women, who accumulate more subcutaneous fat [69]. With COVID-19 severity being higher in men, and also significantly associated with higher visceral fat (but not higher abdominal subcutaneous fat) in our study, it is possible that visceral fat is a main risk factor for disease severity, driven by the fact that the disease appears to be targeting adipose tissue cells [70,71]. We show that image-derived phenotypes are useful biomarkers for identifying at-risk subpopulations. Although the severity estimate in the current study may be lower in post-vaccination populations, our research pinpoints underlining risk factors.

In our analysis, we observed that a smaller iliopsoas volume was associated with a more severe disease outcome. The psoas muscle is a well-recognized marker of frailty and health outcomes [72], with a small psoas muscle area index measured by ultrasound linked to increased mortality in COVID-19 [73]. However, most studies attempting to determine the relationship between muscle mass and COVID-19 outcomes have repurposed chest images obtained during clinical CT/MRI investigations [74–76], whereas our study relies upon standardized research protocols and muscle volumes.

Disease outcome prediction analysis (Table 5) suggested that body composition IDPs from pre-pandemic imaging improved prediction of COVID-19 severity status. We found that abdominal IDPs improved performance accuracy of a model considering only demographic and anthropometric parameters. In addition to established risk factors of BMI, sex, and age, body composition IDPs therefore have prognostic value for COVID-19 disease severity.

A limitation of the study related to the relative time between scans, which ranged between 1.0–7.3 years (3.2 on average), though we adjusted our models for the difference in age between scans. Although this was a prospective study, subjects were not recruited for baseline MRI scanning in anticipation of the COVID-19 pandemic, therefore changes in the body composition could have arisen from changes in lifestyle or disease progression, unrelated to COVID-19. However, our analysis controlled for multiple confounding factors. Moreover, in a previous separate longitudinal study using a larger UKBB cohort, we reported small but significant changes in several tissues and organs [77], although none of these were further altered by COVID-19 in our current study. Thus, the fact that in our previous longitudinal study, we did not observe changes in lung volume further increases confidence in our current findings. A further limitation of the study is that the UKBB population does not include younger people or children, with initial recruitment in 2007 covering participants aged 40 to 69 of age, therefore it is unclear whether our observations can be extrapolated to other age groups.

In conclusion, body composition assessed via MRI and image-derived phenotypes can provide significant insight into the impact of COVID-19 and could help to understand its long-term impact on those suffering its aftermath. Our study showed a significant decrease in lung

volume in SARS-CoV-2 infected cases. We also showed that increased COVID-19 disease severity is associated with smaller iliopsoas muscle volume, higher liver fat as well as higher visceral adipose tissue. Risk estimates of infectious disease severity using MRI-derived measurements of body muscularity and fat can add precision to risk determined by binary assessment of disease diagnosis and may have clinical utility in the future to stratify at-risk populations.

## Supporting information

**S1 Table. Associations of demographic parameters and image-derived phenotypes between baseline and rescan visits.**
(XLSX)

**S2 Table. COVID-19 severity and associations with baseline image-derived phenotypes.**
(XLSX)

**S3 Table. COVID-19 severity and associations with baseline image-derived phenotypes, hospitalization vs non-hospitalization only.**
(XLSX)

**S4 Table. Multivariate model of association for COVID-19 severity.**
(XLSX)

## Acknowledgments

This study has been conducted using the UK Biobank Resource under Application Number 44584. We thank Alan Young and Howard Callen at UK Biobank for facilitating data access, and Amoolya Singh, Chang-Heok Soh and Neha Murad for helpful feedback on the manuscript.

## Author Contributions

**Conceptualization:** Nicolas Basty, Elena P. Sorokin, E. Louise Thomas.

**Data curation:** Nicolas Basty, Elena P. Sorokin, Ramprakash Srinivasan, Brandon Whitcher.

**Formal analysis:** Elena P. Sorokin.

**Funding acquisition:** Jimmy D. Bell.

**Investigation:** Elena P. Sorokin, E. Louise Thomas.

**Methodology:** Nicolas Basty, Elena P. Sorokin, Brandon Whitcher.

**Project administration:** Jimmy D. Bell, Madeleine Cule, E. Louise Thomas.

**Resources:** Ramprakash Srinivasan, Madeleine Cule.

**Software:** Nicolas Basty, Elena P. Sorokin, Marjola Thanaj, Ramprakash Srinivasan, Brandon Whitcher, Madeleine Cule.

**Supervision:** Jimmy D. Bell, E. Louise Thomas.

**Validation:** Marjola Thanaj, Brandon Whitcher.

**Visualization:** Nicolas Basty, Elena P. Sorokin.

**Writing – original draft:** Nicolas Basty, Elena P. Sorokin, Marjola Thanaj, Brandon Whitcher, Jimmy D. Bell, E. Louise Thomas.

**Writing – review & editing:** Nicolas Basty, Elena P. Sorokin, Marjola Thanaj, Ramprakash Srinivasan, Brandon Whitcher, Jimmy D. Bell, Madeleine Cule, E. Louise Thomas.

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
