## [Decision Letter · Decision Letter 0]

10 May 2022

PONE-D-22-06006Abdominal Imaging Associates Body Composition with COVID-19 SeverityPLOS ONE

Dear Dr. Basty,

Thank you for submitting your manuscript to PLOS ONE. After careful consideration, we feel that it has merit but does not fully meet PLOS ONE’s publication criteria as it currently stands. Therefore, we invite you to submit a revised version of the manuscript that addresses the points raised during the review process.

We look forward to receiving your revised manuscript.

Kind regards,

Ivana Isgum

Academic Editor

PLOS ONE

Journal Requirements:

“This work was made possible by the UK Biobank, including staff, funders, and study volunteers. We thank Alan Young and Howard Callen at UK Biobank for facilitating data access, and Amoolya Singh, Chang-Heok Soh and Neha Murad for helpful feedback on the manuscript. This research has been conducted using the UK Biobank Resource under Application Number 44584 and was funded by Calico Life Sciences LLC.”

“This research was funded by Calico Life Sciences LLC.

https://www.calicolabs.com”

Additional Editor Comments (if provided):

The paper has been evaluated by two expert reviewers. Both reviewers recognize the strengths of the study but they have also identified a number of major and minor issues that need to be addressed before the manuscript can be considered for publication. In addition to the points raised by the Reviewers, I would ask the authors to more clearly describe 1) in subsection "Predictive modeling" whether the data was split on the subject level in the cross-validation and 2) Throughout the manuscript what data exactly was used as input in these models (differences between baseline and follow-up vs both time points; this is not very clear everywhere).

Reviewers' comments:

Reviewer's Responses to Questions

**Comments to the Author**

1. Is the manuscript technically sound, and do the data support the conclusions?

Reviewer #1: Yes

Reviewer #2: Yes

2. Has the statistical analysis been performed appropriately and rigorously? 

Reviewer #1: Yes

Reviewer #2: Yes

3. Have the authors made all data underlying the findings in their manuscript fully available?

Reviewer #1: Yes

Reviewer #2: Yes

4. Is the manuscript presented in an intelligible fashion and written in standard English?

Reviewer #1: Yes

Reviewer #2: Yes

5. Review Comments to the Author

Reviewer #1: This study performed a large-scale analysis for abdominal imaging associates body composition with COVID-19 severity. This is timely needed study with comprehensive analysis and decent writing.

Does the author perform a human involved QA process for image processing results. I can imagine some segmentation results are failed.

It would be helpful to illustrate the rationale of using image processing pipeline [37].

Why so many factors have large confidence intervals in Figure 2 and 3?

It seems the trend of results is not consistent between Figure 2 and 3 for statistical significance.

From Figure 4a, the BMI is a much more impactful feature compared with Age. But in Figure 4b, their importance are almost the same. That is somehow contradictory.

Why the visceral fat and liver fat are not used in the Figure 4a?

It seems the top predictors are all fat related features, which would have strong interaction.

Reviewer #2: In this study the authors analyzed data of UK Biobank recruited recovered SARS-CoV-2 positive individuals (n=967) and matched controls (n=913) who were extensively imaged prior to the pandemic and underwent follow-up scanning. They specifically investigated longitudinal changes in body composition, as well as the associations of pre-pandemic image-derived phenotypes with COVID-19 severity. The authors found a decrease in lung volume with SARS-CoV-2 positivity. They also observed that increased visceral adipose tissue and liver fat, and reduced muscle volume, prior to COVID-19, were associated with COVID-19 disease severity. In addition, when training a machine classifier with demographic, anthropometric and imaging traits, they found that visceral fat, liver fat and muscle volume have prognostic value for COVID-19 disease severity beyond the standard demographic and anthropometric measurements. The authors have much expertise in this field of research and the topic addressed by this study is important.

Specific comments:

1. Introduction: When the authors address post-acute sequelae of COVID-19 they should also discuss the findings of very important studies that investigated this question (Nature. 2021 Jun;594(7862):259-264; Nat Med. 2022 Mar;28(3):583-590; Lancet Diabetes Endocrinol. 2022 Mar 21:S2213-8587(22)00044-4. doi: 10.1016/S2213-8587(22)00044-4).

2. Introduction: The authors should also discuss how the COVID-19 pandemic may generally impact on the cardiometabolic risk and on obesity (Nat Rev Endocrinol. 2021 Mar;17(3):135-149).

3. Impaired cardiometabolic health and obesity are thought to promote SARS-CoV-2 vaccine-breakthrough infections (Nat Rev Endocrinol. 2022 Feb;18(2):75-76). Do the authors have data to address this important point? Otherwise, they should at least carefully discuss this point.

4. When the authors address the relationship of obesity with severity of COVID-19, they should also discuss the important findings from the QResearch database of general practices in England, UK study (Lancet Diabetes Endocrinol. 2021 Jun;9(6):350-359), which carefully investigated the relationship of obesity, under the aspects of different comorbidities and age, with the severity of COVID-19.

5. On what basis were specifically the 1,880 participants, who attended the first imaging visit, selected and recruited to the new re-imaging study? It appears as if these 1,880 participants are the cases and controls. So, what was the initial number of subjects with a positive SARS-CoV-2 test result that was asked to attended the second imaging visit?

6. The authors assessed longitudinal effects of SARS-CoV-2 infection, as well as the severity of COVID-19 based on pre-pandemic imaging alone. Does this mean that from the 1,955 participants mentioned in the result section, data of those wo died and of those who refused to participate in the second imaging study, were analyzed for cross-sectional relationships?

7. Table 1: For smoking, alcohol consumption and selected diseases reported, the authors indicate that the percentage of the subjects is reported. However, from the data of that table the number cannot be percentages.

8. Table 1: If the percentage of subjects having NAFLD amounts to 70-75, why was the percentage of patients with type 2 diabetes so low (2-3 percent). Considering the prevalence of NAFLD in patients with type 2 diabetes (Lancet Diabetes Endocrinol. 2022 Apr;10(4):284-296), these numbers should be different.

9. Unexpectedly, the authors only observed a significant decrease in lung volume, but no change in any other examined image-derived phenotypes. May it be possible that most of the participants, who attended the second imaging study, had relatively mild COVID-19 and patients with a more severe course of the disease refused to participate in the second exam?

6. PLOS authors have the option to publish the peer review history of their article (what does this mean?). If published, this will include your full peer review and any attached files.

Reviewer #1: No

Reviewer #2: No

---

## [Author Response · Author response to Decision Letter 0]

19 Jul 2022

We have edited the manuscript accordingly and made changes to match the house style including size and bold font section heading, numbered equations, replaced Figure with Fig in the text, moved table titles to the top of the tables, renamed supplementary tables to S1 Table throughout text, and moved the supporting information to the end of the manuscript after the reference list.

We have clarified the text detailing the ethics and consent covering this study in the study design section.

“This work was made possible by the UK Biobank, including staff, funders, and study volunteers. We thank Alan Young and Howard Callen at UK Biobank for facilitating data access, and Amoolya Singh, Chang-Heok Soh and Neha Murad for helpful feedback on the manuscript. This research has been conducted using the UK Biobank Resource under Application Number 44584 and was funded by Calico Life Sciences LLC.”

“This research was funded by Calico Life Sciences LLC.

https://www.calicolabs.com”

We have removed reference to funding from the acknowledgement. The corrected funding statement is: 

This research was funded by Calico Life Sciences LLC. https://calicolabs.com

Our research was conducted using UK Biobank data. Under the standard UK Biobank data sharing agreement, researchers are unable to directly share data they obtain from the biobank with others. However, all of the the the data generated and used in this paper are available directly from the UK Biobank upon request at https://www.ukbiobank.ac.uk and we have included this in the data availability statement.

Additional Editor Comments (if provided):

The paper has been evaluated by two expert reviewers. Both reviewers recognize the strengths of the study but they have also identified a number of major and minor issues that need to be addressed before the manuscript can be considered for publication. In addition to the points raised by the Reviewers, I would ask the authors to more clearly describe 1) in subsection "Predictive modeling" whether the data was split on the subject level in the cross-validation and 2) Throughout the manuscript what data exactly was used as input in these models (differences between baseline and follow-up vs both time points; this is not very clear everywhere).

We would like to thank the editor for this comment, we have addressed this point in the methods and the paper has clearly benefited from this added clarity. We have split the methods into subsections, clearly differentiating the longitudinal analysis and the baseline-only analyses. The predictive modeling was performed on the baseline data only, as we tried to predict disease severity from image-derived body composition before the pandemic. The data were not paired at subject level in the predictive analysis as only one time point was used, and therefore could not be split.

Reviewer #1: This study performed a large-scale analysis for abdominal imaging associates body composition with COVID-19 severity. This is timely needed study with comprehensive analysis and decent writing.

Does the author perform a human involved QA process for image processing results. I can imagine some segmentation results are failed.

This is a good point to raise, quality control with big data has to be very stringent, and we did indeed perform extensive QA by visually checking the fifty largest and fifty smallest volumes, as well as another 50 random for each structure. The QA protocol is outlined in [37], however we have added some additional details into the manuscript to explain this more clearly. 

It would be helpful to illustrate the rationale of using image processing pipeline [37].

We thank the reviewer for pointing this out. The preprocessing steps, segmentation models for individual organs and the quantitative analysis of PDFF (proton density fat fraction) and R2* have been applied to approximately 38,000 scans from the UK Biobank in Liu et al. (2021). We have added text in the “Image-derived phenotypes” section that provides the rationale. 

Why so many factors have large confidence intervals in Figure 2 and 3?

The variables affected by large confidence intervals in Figure 3 (smoking and alcohol) are self-reported, and therefore have more variability compared to quantitative values; there could also be a bias in answering those questions, given their negative connotation. For Figure 3, the confidence intervals are also quite large for sex as well as ethnicity and imaging centre. We believe this is due to the uneven distribution of severe cases as for example there are about three times more severe cases in men in that particular dataset.

It seems the trend of results is not consistent between Figure 2 and 3 for statistical significance.

We thank the reviewer for this remark, the plots are quite similar, but the outcome variable is different. Figure 2 summarises the changes between the two timepoints between cases and controls, whereas Figure 3 is about investigating the associations between image-derived phenotypes (IDPs) and severity. We have made this more explicit in the text and captions.

From Figure 4a, the BMI is a much more impactful feature compared with Age. But in Figure 4b, their importance are almost the same. That is somehow contradictory.

Thank you for raising this point.. Figure 4A are the ROC curves for the three different models (Age+Sex, Age+Sex+Height+BMI, Age+Sex+BMI+Height+IDPs) whereas figure 4B is the importance of the random forest features for the last model only. We have changed the caption to reflect this.

Why the visceral fat and liver fat are not used in the Figure 4a?

Thank you for this question, which falls into the same category as the one above. We did use visceral fat and liver fat in figure 4A, the green line represents the model with Age+Sex+BMI+IDPs, where IDPs are the image-derived phenotypes (visceral fat and liver fat, iliopsoas muscle volume). We have amended the figure caption accordingly to remove any ambiguity.

It seems the top predictors are all fat related features, which would have strong interaction.

Yes, we agree that several of the key variables are likely to correlate with each other. We included a regularisation term in the regression model, using the LASSO (Least Absolute Shrinkage and Selection Operator), to perform variable selection and produce a more parsimonious and interpretable result.

Reviewer #2: In this study the authors analyzed data of UK Biobank recruited recovered SARS-CoV-2 positive individuals (n=967) and matched controls (n=913) who were extensively imaged prior to the pandemic and underwent follow-up scanning. They specifically investigated longitudinal changes in body composition, as well as the associations of pre-pandemic image-derived phenotypes with COVID-19 severity. The authors found a decrease in lung volume with SARS-CoV-2 positivity. They also observed that increased visceral adipose tissue and liver fat, and reduced muscle volume, prior to COVID-19, were associated with COVID-19 disease severity. In addition, when training a machine classifier with demographic, anthropometric and imaging traits, they found that visceral fat, liver fat and muscle volume have prognostic value for COVID-19 disease severity beyond the standard demographic and anthropometric measurements. The authors have much expertise in this field of research and the topic addressed by this study is important.

Specific comments:

1. Introduction: When the authors address post-acute sequelae of COVID-19 they should also discuss the findings of very important studies that investigated this question (Nature. 2021 Jun;594(7862):259-264; Nat Med. 2022 Mar;28(3):583-590; Lancet Diabetes Endocrinol. 2022 Mar 21:S2213-8587(22)00044-4. doi: 10.1016/S2213-8587(22)00044-4).

We thank the reviewer for highlighting these papers. Given the extraordinary number of relevant papers that have been published since the pandemic began it was not possible to include all of the references we would have liked. At the time of writing we focussed on the most organ specific publications most directly relevant to this manuscript. We however have expanded our Introduction to include the suggested texts and the paper clearly benefits from this addition.

2. Introduction: The authors should also discuss how the COVID-19 pandemic may generally impact on the cardiometabolic risk and on obesity (Nat Rev Endocrinol. 2021 Mar;17(3):135-149).

We have expanded the part of the discussion where we describe how chronic comorbidities such as obesity, impaired metabolic health, and diabetes are known to be at higher risk of severe disease and poorer prognosis following COVID-19 infection to include this excellent review by Stefan et al.

3. Impaired cardiometabolic health and obesity are thought to promote SARS-CoV-2 vaccine-breakthrough infections (Nat Rev Endocrinol. 2022 Feb;18(2):75-76). Do the authors have data to address this important point? Otherwise, they should at least carefully discuss this point.

The referee raises an important point. Unfortunately we do not have any specific data available to us within this cohort regarding vaccine-breakthrough infections however we have included a reference to this in the introduction. 

4. When the authors address the relationship of obesity with severity of COVID-19, they should also discuss the important findings from the QResearch database of general practices in England, UK study (Lancet Diabetes Endocrinol. 2021 Jun;9(6):350-359), which carefully investigated the relationship of obesity, under the aspects of different comorbidities and age, with the severity of COVID-19.

We have added this reference to the text. 

5. On what basis were specifically the 1,880 participants, who attended the first imaging visit, selected and recruited to the new re-imaging study? It appears as if these 1,880 participants are the cases and controls. So, what was the initial number of subjects with a positive SARS-CoV-2 test result that was asked to attended the second imaging visit? 

The UK Biobank rescanning of participants for the COVID study initially aimed to recruit 2,000 participants, the inclusion criteria are listed in detail here: https://biobank.ndph.ox.ac.uk/showcase/showcase/docs/casecontrol_covidimaging.pdf, but in brief participants needed to: have already attended pre-pandemia imaging assessment at one of the three imaging sites, live within 60km of the clinic (COVID travel restrictions were still in place during the rescanning study) and had high-quality UK Biobank MRI images available from their first imaging visit. These criteria are in our section “study design” on page 6. At the time of preparing this manuscript, a total of 1,880 suitable datasets were available for the study.

6. The authors assessed longitudinal effects of SARS-CoV-2 infection, as well as the severity of COVID-19 based on pre-pandemic imaging alone. Does this mean that from the 1,955 participants mentioned in the result section, data of those wo died and of those who refused to participate in the second imaging study, were analyzed for cross-sectional relationships?

Data from individuals from only one imaging time point at baseline was used only in the severity analysis (Tables 3 and 4)

7. Table 1: For smoking, alcohol consumption and selected diseases reported, the authors indicate that the percentage of the subjects is reported. However, from the data of that table the number cannot be percentages.

We have re-made the table to give percentages rather than fractions for categorical variables for easier reading.

8. Table 1: If the percentage of subjects having NAFLD amounts to 70-75, why was the percentage of patients with type 2 diabetes so low (2-3 percent). Considering the prevalence of NAFLD in patients with type 2 diabetes (Lancet Diabetes Endocrinol. 2022 Apr;10(4):284-296), these numbers should be different.

Thanks to the reviewer. We identified a bug in ascertainment of NAFLD cases, which we have now corrected. In fact, there is only a single participant with NAFLD diagnosed via tertiary care records (defined via ICD10 codes K74.0, K74.1, K74.2, K74.6, K76.0). We have therefore removed this from our analysis as we are not powered to detect differences in incidence or disease severity. To address the poor ascertainment of diabetes from tertiary care records, we have updated our analysis to use an expanded definition of diabetes (UK Biobank Fields 130708 and 130706 for T2D and T1D respectively) which incorporates primary care and self-report as well as tertiary care records. This affects table 1, table 3. We have removed supplementary table 2 as it was the same as T3.

9. Unexpectedly, the authors only observed a significant decrease in lung volume, but no change in any other examined image-derived phenotypes. May it be possible that most of the participants, who attended the second imaging study, had relatively mild COVID-19 and patients with a more severe course of the disease refused to participate in the second exam?

The referee raises an interesting point, and we too had expected to observe more significant changes in other organs given some reports in the literature, although many of these reports are from subjects with very severe disease. The majority of the UKBB subjects invited for a second scan and who had tested positive for COVID-19, had relatively mild disease (96% mild vs 4% severe). We do not have any information whether subjects with more severe disease were less likely to attend for follow-up scanning, but were reassured that the response rates in the COVID and control groups for repeat UKBB scanning were very similar, with a 58% response rate from participants who had COVID-19 and a 57% response rate from participants who have not been infected. It is however a valid point that there may have been some self-filtering where participants who had very severe COVID-19 (or long-COVID effects) did not want to undergo a second scanning exam; we have added this as a caveat to the study.

Regarding the funding, we have removed the separate entry we initially had in the manuscript. The funders had no role in study design, data collection and analysis, decision to publish, or preparation of the manuscript.

---

## [Decision Letter · Decision Letter 1]

29 Sep 2022

PONE-D-22-06006R1Abdominal Imaging Associates Body Composition with COVID-19 SeverityPLOS ONE

Dear Dr. Basty,

Thank you for submitting your manuscript to PLOS ONE. After careful consideration, we feel that it has merit but does not fully meet PLOS ONE’s publication criteria as it currently stands. Therefore, we invite you to submit a revised version of the manuscript that addresses the points raised during the review process.

We look forward to receiving your revised manuscript.

Kind regards,

Ivana Isgum

Academic Editor

PLOS ONE

Journal Requirements:

Additional Editor Comments (if provided):

The authors have performed a careful revision and addressed the comments raised in the review. Nevertheless, Reviewer 3 identified a few issues that I agree with. I would ask the authors to address the first two comments and to consider changing the table for easier interpretation of the results.

Reviewers' comments:

Reviewer's Responses to Questions

**Comments to the Author**

1. If the authors have adequately addressed your comments raised in a previous round of review and you feel that this manuscript is now acceptable for publication, you may indicate that here to bypass the “Comments to the Author” section, enter your conflict of interest statement in the “Confidential to Editor” section, and submit your "Accept" recommendation.

Reviewer #1: All comments have been addressed

Reviewer #2: All comments have been addressed

Reviewer #3: (No Response)

2. Is the manuscript technically sound, and do the data support the conclusions?

Reviewer #1: Yes

Reviewer #2: Yes

Reviewer #3: Yes

3. Has the statistical analysis been performed appropriately and rigorously? 

Reviewer #1: Yes

Reviewer #2: Yes

Reviewer #3: Yes

4. Have the authors made all data underlying the findings in their manuscript fully available?

Reviewer #1: Yes

Reviewer #2: Yes

Reviewer #3: No

5. Is the manuscript presented in an intelligible fashion and written in standard English?

Reviewer #1: Yes

Reviewer #2: Yes

Reviewer #3: Yes

6. Review Comments to the Author

Reviewer #1: The quality of the paper has been substantially improved from this round of review. All my concerns have been addressed.

Reviewer #2: The authors have very well addressed the comments and provide very interesting information about the impact of SARS-CoV-2 infection on abdominal organ health, body composition and lung volume.

Reviewer #3: This is the first time I have read this paper, not having been part of the previous review round(s). The paper is well-structured and the text is mostly clear, although it is quite verbose. I have primarily focused on the descriptions of the experimental setting, the statistics and the presentation of the results. My specific comments are listed below.

- In the Longitudinal analysis section, the authors state “(delta age)^2 was calculated as the difference of the squared age at rescanning and the squared age at baseline”. This is notably different from the formula (age_rescan – age_baseline)^2, from this same section. By the procedure in the text, (delta age)^2 for ages 60 and 62 would be 62*62-60*60=244, whereas by the procedure described in the formula, (delta age)^2 would be 4. It is unclear which of these results the authors have actually used.

- In the section “Image-derived phenotypes”, the authors describe a data validation procedure: visually inspecting a subset of 150 scans (primarily outliers in terms of volume) to ensure the used pipeline is robust. I may have missed it, but I do not believe they describe how possible mistakes were handled. Were all 150 validated scans perfect? If not, were they manually corrected before being used in the analysis presented in this work? This should be clear in the manuscript.

- The tables are not pleasant to parse. Many depicted values have more significant figures than would be relevant, resulting in visual clutter. Currently, they seem like tables meant to be copy-pasted into a computer program, rather than tables meant to be readable by humans. I urge the authors to revisit all tables to improve readability. For example, the FDR column in Table 1 can be deleted as it does not add any information here and the ASAT row in Table 2 should be rounded to milliliters.

7. PLOS authors have the option to publish the peer review history of their article (what does this mean?). If published, this will include your full peer review and any attached files.

Reviewer #1: No

Reviewer #2: **Yes: **Norbert Stefan

Reviewer #3: No

---

## [Author Response · Author response to Decision Letter 1]

12 Oct 2022

Journal Requirements:

We have double checked each article individually in the reference list. No articles have been retracted. However, several that were at the preprint stage have since been published and we have updated the references accordingly:

 34. Scalsky RJ, Chen Y-J, Desai K, O’Connell JR, Perry JA, Hong CC. Baseline cardiometabolic profiles and SARS-CoV-2 infection in the UK Biobank. PLoS One. 2021;16: e0248602.

37. Douaud G, Lee S, Alfaro-Almagro F, Arthofer C, Wang C, McCarthy P, et al. SARS-CoV-2 is associated with changes in brain structure in UK Biobank. Nature. 2022;604: 697–707.

 38. Sattar N, Ho FK, Gill JM, Ghouri N, Gray SR, Celis-Morales CA, et al. BMI and future risk for COVID-19 infection and death across sex, age and ethnicity: Preliminary findings from UK biobank. Diabetes Metab Syndr. 2020;14: 1149–1151.

 41. Willette AA, Willette SA, Wang Q, Pappas C, Klinedinst BS, Le S, et al. Using machine learning to predict COVID-19 infection and severity risk among 4510 aged adults: a UK Biobank cohort study. Sci Rep. 2022;12: 7736.

Additional Editor Comments (if provided):

The authors have performed a careful revision and addressed the comments raised in the review. Nevertheless, Reviewer 3 identified a few issues that I agree with. I would ask the authors to address the first two comments and to consider changing the table for easier interpretation of the results.

We thank the editor for the additional comments and finding an additional reviewer.

Review Comments to the Author

Reviewer #1: The quality of the paper has been substantially improved from this round of review. All my concerns have been addressed.

We thank reviewer #1 for taking the time to review our manuscript and revisions.

Reviewer #2: The authors have very well addressed the comments and provide very interesting information about the impact of SARS-CoV-2 infection on abdominal organ health, body composition and lung volume.

We thank reviewer #2 for taking the time to review our manuscript and revisions.

Reviewer #3: This is the first time I have read this paper, not having been part of the previous review round(s). The paper is well-structured and the text is mostly clear, although it is quite verbose. I have primarily focused on the descriptions of the experimental setting, the statistics and the presentation of the results. My specific comments are listed below.

We thank reviewer #3 for taking the time to review our manuscript, and agree with the comments intended to make the paper easier to read.

- In the Longitudinal analysis section, the authors state “(delta age)^2 was calculated as the difference of the squared age at rescanning and the squared age at baseline”. This is notably different from the formula (age_rescan – age_baseline)^2, from this same section. By the procedure in the text, (delta age)^2 for ages 60 and 62 would be 62*62-60*60=244, whereas by the procedure described in the formula, (delta age)^2 would be 4. It is unclear which of these results the authors have actually used.

Thank-you for pointing out this typo. The formula was incorrect and has been modified to match the text.

- In the section “Image-derived phenotypes”, the authors describe a data validation procedure: visually inspecting a subset of 150 scans (primarily outliers in terms of volume) to ensure the used pipeline is robust. I may have missed it, but I do not believe they describe how possible mistakes were handled. Were all 150 validated scans perfect? If not, were they manually corrected before being used in the analysis presented in this work? This should be clear in the manuscript.

We have added text clarifying that no mistakes were found during the manual visual inspection of the data and it was therefore non necessary to undertake any corrective manual editing. 

- The tables are not pleasant to parse. Many depicted values have more significant figures than would be relevant, resulting in visual clutter. Currently, they seem like tables meant to be copy-pasted into a computer program, rather than tables meant to be readable by humans. I urge the authors to revisit all tables to improve readability. For example, the FDR column in Table 1 can be deleted as it does not add any information here and the ASAT row in Table 2 should be rounded to milliliters.

We agree with the reviewer that the tables benefit from decluttering and have rounded volumes to millilitres. We removed the FDR column from Table 1, and removed the FDR and Statistic columns from Table 3. Asterisks have been added to p-values that are statistically significant in both tables, with additional text provided for explanation.

---

## [Decision Letter · Decision Letter 2]

10 Jan 2023

PONE-D-22-06006R2Abdominal Imaging Associates Body Composition with COVID-19 SeverityPLOS ONE

Dear Dr. Basty,

Thank you for submitting your manuscript to PLOS ONE. After careful consideration, we feel that it has merit but does not fully meet PLOS ONE’s publication criteria as it currently stands. Therefore, we invite you to submit a revised version of the manuscript that addresses the points raised during the review process.

#reviewer 3I commend the authors for their work in improving the manuscript. However, I am quite surprised to hear that there were no errors at all in any of the visually inspected segmentations. Somewhat suspect is that the rendered lung segmentation in figure 1 seems to contain a disconnected left bronchus, which would generally be considered a segmentation error. The authors should define what was considered acceptable and what was considered an error during this quality control step.

The tables look a lot better now. A final (minor) suggestion would be to change any p-values larger than 0.10 to decimal notation in Table 3 to make it easier to differentiate large from small numbers at a glance (as 5.53E-01 takes half a second to decipher, versus a few milliseconds for 0.55).Please ensure that your decision is justified on PLOS ONE’s publication criteria and not, for example, on novelty or perceived impact.

We look forward to receiving your revised manuscript.

Kind regards,

Ying-Mei Feng

Academic Editor

PLOS ONE

Journal Requirements:

Reviewers' comments:

Reviewer's Responses to Questions

**Comments to the Author**

1. If the authors have adequately addressed your comments raised in a previous round of review and you feel that this manuscript is now acceptable for publication, you may indicate that here to bypass the “Comments to the Author” section, enter your conflict of interest statement in the “Confidential to Editor” section, and submit your "Accept" recommendation.

Reviewer #1: (No Response)

Reviewer #3: (No Response)

2. Is the manuscript technically sound, and do the data support the conclusions?

Reviewer #1: (No Response)

Reviewer #3: Yes

3. Has the statistical analysis been performed appropriately and rigorously? 

Reviewer #1: (No Response)

Reviewer #3: Yes

4. Have the authors made all data underlying the findings in their manuscript fully available?

Reviewer #1: (No Response)

Reviewer #3: No

5. Is the manuscript presented in an intelligible fashion and written in standard English?

Reviewer #1: (No Response)

Reviewer #3: Yes

6. Review Comments to the Author

Reviewer #1: (No Response)

Reviewer #3: I commend the authors for their work in improving the manuscript. However, I am quite surprised to hear that there were no errors at all in any of the visually inspected segmentations. Somewhat suspect is that the rendered lung segmentation in figure 1 seems to contain a disconnected left bronchus, which would generally be considered a segmentation error. The authors should define what was considered acceptable and what was considered an error during this quality control step.

The tables look a lot better now. A final (minor) suggestion would be to change any p-values larger than 0.10 to decimal notation in Table 3 to make it easier to differentiate large from small numbers at a glance (as 5.53E-01 takes half a second to decipher, versus a few milliseconds for 0.55).

7. PLOS authors have the option to publish the peer review history of their article (what does this mean?). If published, this will include your full peer review and any attached files.

Reviewer #1: No

Reviewer #3: No

---

## [Author Response · Author response to Decision Letter 2]

24 Jan 2023

#reviewer 3

I commend the authors for their work in improving the manuscript. However, I am quite surprised to hear that there were no errors at all in any of the visually inspected segmentations. Somewhat suspect is that the rendered lung segmentation in figure 1 seems to contain a disconnected left bronchus, which would generally be considered a segmentation error. The authors should define what was considered acceptable and what was considered an error during this quality control step.

We thank the Reviewer for taking the time to review the latest iteration of our paper. We agree that the wording of the statement is too generous and we have amended this statement that was in the text:

We found no errors in the visual inspections and did not perform any manual editing.

To as follows:

During visual quality control of the segmentations, no catastrophic failures were observed and we did not perform any manual editing. We defined a catastrophic failure to be when segmented organs had major components missing or components that did not belong to the structure added (i.e. over or under segmentation). This could be due to underlying data errors such as fat-water swaps, or where the neural network did or did not highlight part of the organ even though the underlying image intensities are free of ambiguities. 

The tables look a lot better now. A final (minor) suggestion would be to change any p-values larger than 0.10 to decimal notation in Table 3 to make it easier to differentiate large from small numbers at a glance (as 5.53E-01 takes half a second to decipher, versus a few milliseconds for 0.55).

Thank you for this comment which improves the readability of the tables. We have amended the tables as suggested.

---

## [Decision Letter · Decision Letter 3]

12 Mar 2023

Abdominal Imaging Associates Body Composition with COVID-19 Severity

PONE-D-22-06006R3

Dear Dr. Basty,

We’re pleased to inform you that your manuscript has been judged scientifically suitable for publication and will be formally accepted for publication once it meets all outstanding technical requirements.

Kind regards,

Jun Mori

Academic Editor

PLOS ONE

Additional Editor Comments (optional):

Reviewers' comments:

Reviewer's Responses to Questions

**Comments to the Author**

1. If the authors have adequately addressed your comments raised in a previous round of review and you feel that this manuscript is now acceptable for publication, you may indicate that here to bypass the “Comments to the Author” section, enter your conflict of interest statement in the “Confidential to Editor” section, and submit your "Accept" recommendation.

Reviewer #3: All comments have been addressed

2. Is the manuscript technically sound, and do the data support the conclusions?

Reviewer #1: (No Response)

Reviewer #3: (No Response)

3. Has the statistical analysis been performed appropriately and rigorously? 

Reviewer #1: (No Response)

Reviewer #3: (No Response)

4. Have the authors made all data underlying the findings in their manuscript fully available?

Reviewer #1: (No Response)

Reviewer #3: (No Response)

5. Is the manuscript presented in an intelligible fashion and written in standard English?

Reviewer #1: (No Response)

Reviewer #3: (No Response)

6. Review Comments to the Author

Reviewer #1: I have recommended acceptance in the last round of review. Please do not send it to me again since that was my final decision.

Reviewer #3: (No Response)

7. PLOS authors have the option to publish the peer review history of their article (what does this mean?). If published, this will include your full peer review and any attached files.

Reviewer #1: No

Reviewer #3: No

---

## [Editor Report · Acceptance letter]

4 Apr 2023

PONE-D-22-06006R3 

Abdominal Imaging Associates Body Composition with COVID-19 Severity 

Dear Dr. Basty:

I'm pleased to inform you that your manuscript has been deemed suitable for publication in PLOS ONE. Congratulations! Your manuscript is now with our production department. 

Kind regards, 

on behalf of

Dr. Jun Mori 

Academic Editor

PLOS ONE